# Nutritional Intervention to Prevent the Functional Decline in Community-Dwelling Older Adults: A Systematic Review

**DOI:** 10.3390/nu12092820

**Published:** 2020-09-15

**Authors:** Julie Mareschal, Laurence Genton, Tinh-Hai Collet, Christophe Graf

**Affiliations:** 1Clinical Nutrition, Geneva University Hospital, 1211 Geneva, Switzerland; julie.mareschal@hcuge.ch (J.M.); laurence.genton@hcuge.ch (L.G.); tinh-hai.collet@hcuge.ch (T.-H.C.); 2Department of Rehabilitation and Geriatrics, Geneva University Hospital, 1211 Geneva, Switzerland

**Keywords:** muscle mass, muscle strength, physical performance, sarcopenia, hospital admission, elderly

## Abstract

Aging is a global public health concern. From the age of 50, muscle mass, muscle strength and physical performance tend to decline. Sarcopenia and frailty are frequent in community-dwelling older adults and are associated with negative outcomes such as physical disability and mortality. Therefore, the identification of therapeutic strategies to prevent and fight sarcopenia and frailty is of great interest. This systematic review aims to summarize the impact of nutritional interventions alone or combined with other treatment(s) in older community-dwelling adults on (1) the three indicators of sarcopenia, i.e., muscle mass, muscle strength and physical performance; and (2) the hospitalization and readmission rates. The literature search was performed on Medline and included studies published between January 2010 and June 2020. We included randomized controlled trials of nutritional intervention alone or combined with other treatment(s) in community-living subjects aged 65 or older. In total, 28 articles were retained in the final analysis. This systematic review highlights the importance of a multimodal approach, including at least a combined nutritional and exercise intervention, to improve muscle mass, muscle strength and physical performance, in community-dwelling older adults but especially in frail and sarcopenic subjects. Regarding hospitalization and readmission rate, data were limited and inconclusive. Future studies should continue to investigate the effects of such interventions in this population.

## 1. Introduction

Aging is a global public health concern. According to the World Health Organization (WHO), by the end of the decade, the number of people aged 60 and over will have increased worldwide by 34%, from 1 billion in 2019 to 1.4 billion [1]. Compared to 2019, this population will have more than doubled to reach 2.1 billion by 2050. The aging process is associated with physical, cognitive and social changes affecting morbidity and mortality [2]. Particularly, muscle mass, muscle strength and physical performance tend to decline [3,4]. From the age of 50, an individual loses 1% to 2% of leg muscle mass per year and 1.5% to 5% of leg strength [3].

Sarcopenia is defined by the European Working Group on Sarcopenia in Older People (EWGSOP) as an association between two indicators: low muscle mass, low muscle strength or low physical performance [5]. In community-dwelling older adults, the prevalence of sarcopenia ranges from 0.2% to 20% according to the used parameters and the population, and increases with age [6,7,8]. Sarcopenia is associated with negative outcomes, such as physical disability [9], limitations in activities of daily living [10], falls [11], hospitalizations and readmissions [12], loss of autonomy with a need for home healthcare or nursing home placement [13,14] and mortality [9,15,16]. The health economics burden of sarcopenia is estimated to be over $18 billion a year in the United States [17,18]. Frailty is a clinical syndrome similar to sarcopenia. It includes at least three of the following criteria: unintentional weight loss, self-reported exhaustion, weakness, slow walking, and low physical activity [19]. Multiple therapeutic strategies have been investigated to prevent and fight sarcopenia and frailty in older adults. Nutrient intake plays a central role in the development and maintenance of muscle mass and strength and is therefore a key element [20].

This systematic review aims to summarize the impact of nutritional interventions, defined as calorie and/or protein supplementation, alone or combined with other treatments in older community-dwelling adults on (1) three indicators of sarcopenia: muscle mass, muscle strength, and physical performance, and (2) hospitalization and readmission rates.

## 2. Materials and Methods

This systematic review was carried out according the Preferred Reporting Items for Systematic Reviews and Meta-Analyses (PRISMA) guidelines [21].

### 2.1. Study Eligibility Criteria

We included all primary source randomized controlled trials in the English language published within the last 10 years (1 January 2010–30 June 2020) which met all the following eligibility criteria. Population: older adults in the community, mean age ≥ 65 years, mono-morbid specific population. Intervention: nutritional intervention, defined as calorie and/or protein supplementation, alone or combined with other treatment(s) (i.e., physical activity = resistance, endurance, balance, flexibility, and electrostimulation, cognitive training, and androgen therapy), minimum 1-month follow-up, exclusion of nutritional advice or counseling as the sole intervention of interest. Comparison: placebo control group or other nutritional intervention or other treatment(s) or combined nutritional intervention with other treatment(s). Outcomes: muscle mass, muscle strength, physical performance, and hospitalization and readmission rates.

### 2.2. Study Identification

The MEDLINE electronic database (Pubmed) was used to identify eligible articles. The first search was performed in April 2020 and updated on 6 July 2020. The search strategy was discussed between the authors and defined as follows: (“Elderly” OR “Aged” OR “Geriatric*” OR “Frail” OR “Old” OR “Older”) AND (“Community*” OR “Outpatient*” OR “Home-based” OR “Out patient*”) AND (“Nutrition*” OR “Dietary supplement*” OR “Dietary protein*” OR “Protein” OR “Amino acid*” OR “Caloric*”) AND (“Hospitalization*” OR “Hospitalisation*” OR “Readmission*” OR “Function*” OR “Performance” OR “Activities of daily living” OR “Muscle*” OR “Strength*” OR “Fat-free mass” OR “Lean body mass” OR “Lean tissue mass” or “Fall*”) AND (“Controlled trial*” OR “Random*”).

### 2.3. Study Selection

One author (J.M.) assessed the titles and abstracts of the retrieved articles for eligibility after the literature search. When an abstract or a title met the inclusion criteria, the full text was reviewed to evaluate the eligibility and inclusion of the article in the systematic review. Four articles were also manually selected from the references of the selected articles. The final decision to include the articles in the systematic review was made after reaching a consensus among all authors. The study selection process is detailed in Figure 1.

### 2.4. Data Collection and Study Quality

The following data from selected articles were extracted in a standard form: population, age, sample size, intervention (duration and description), assessed outcomes, results, and major limitations. The outcomes were muscle mass evaluated by bioelectrical impedance analysis (BIA), dual-energy X-ray absorptiometry (DXA) or magnetic resonance imaging (MRI), muscle strength (e.g., handgrip strength, one repetition maximum, isokinetic strength, knee extension strength), physical performance (e.g., Short Physical Performance Battery (SPPB) score, six-minute walk test, timed up and go, one-minute sit-to-stand test), and hospitalization and readmission rates. The results were the impact of the nutritional intervention alone or combined with other treatment(s) on the different outcomes. The major limitations of each study were also reported.

A score, derived from the checklist proposed by Downs and Black [22], was used to evaluate the quality of the studies. This score includes 27 items checklist. We simplified the scoring of item number 27 and rated a study as one if a power calculation was performed and zero if not. Accordingly, the maximum score was 28 instead of 32. Quality levels were given as follows: excellent (score 24–28); good (score 19–23); fair (score 14–18); and poor (score < 14), as previously reported by O’Connor et al [23].

## 3. Results

### 3.1. Study Selection

The flow diagram (Figure 1) describes the studies selection process. The literature search identified 1275 articles and 4 additional articles were found manually through the references of these articles. A total of 1279 abstracts and titles were screened and 69 full-text articles were assessed for eligibility. Finally, 28 articles were included in the review.

### 3.2. Study Characteristics

The study characteristics are presented in Table 1, Table 2, Table 3 and Table 4. The population was heterogeneous with healthy, mobility-limited, frail, and sarcopenic subjects. Few studies tested a nutritional intervention as a sole intervention of interest (*n* = 5, Table 1). The other studies combined the nutritional intervention with other approaches, such as physical activity or exercise (*n* = 18, Table 2) and testosterone therapy (*n* = 2, Table 3). Finally, three studies had multimodal interventions: nutritional and physical activity interventions and/or cognitive and/or testosterone therapies (Table 4). The modalities of intervention differed between the majority of studies. The duration of interventions ranged from 4 weeks to 2 years. The methods used to assess the different outcomes varied and may not have been standardized.

### 3.3. Risk of Bias within Studies

The checklist proposed by Downs and Black [22] was used to assess the study quality (right-most column of Table 1, Table 2, Table 3 and Table 4). Most studies were classified as “good” (23 out of 28), while 2 studies were evaluated as “excellent” and 3 studies as “fair”.

### 3.4. Main Findings

#### 3.4.1. Unimodal Nutritional Interventions

In healthy older adults, the effects of nutritional intervention on muscle mass, muscle strength, and physical performance were divergent. In a small sample, Ellis et al. showed that a 6-month amino acid supplementation with beta-hydroxy-beta-methylbutyrate (HMB) improved lean body mass (52.9 ± 11.9 vs. 48.4 ± 11.4 kg, *p* = 0.036) and 12-step timed stair climb (4.86 ± 1.66 vs. 4.61 ± 0.80 s, *p* = 0.016) [24]. Zhu et al. tested the effect of whey protein supplementation (30 g/day) during two years, in Australian subjects with baseline protein intake above 0.75 g/kg/day [25]. They reported no change in appendicular skeletal muscle mass (ASMM), muscle strength, and physical performance. Finally, similar results were reported by Ottestad et al. who failed to demonstrate improvement in these outcomes after a 12-week protein supplementation (40 g/day) [26]. Nevertheless, a high dropout rate was documented in both treatment arms.

In Korea, Kim et al. evaluated the benefits of a 12-week oral nutritional supplementation (200 kcal, 12.5 g of proteins, 2 ×/day) in frail older adults on handgrip strength and physical performance [27]. They only found a significant improvement of 1.1 s in the timed-up and go test (vs. −0.9 s in controls, *p* = 0.039) while the short physical performance battery (SPPB) score remained stable in the intervention group and decreased by 1 point in the control group (*p* = 0.038). The major limitations of this study were the high dropout rate in the oral nutritional supplementation group and the absence of placebo in the control group.

In a large sample of European sarcopenic subjects, a 13-week leucine-enriched whey protein and vitamin D supplement (20 g of whey protein, 3 g of leucine and 800 IU vitamin D), consumed twice daily, significantly improved ASSM and 5-time sit-to-stand test compared to an isocaloric placebo [28]. However, no differences in handgrip strength, SPPB, gait speed and balance scores were demonstrated. The high dropout rate is again a major limitation of this study. Moreover, the definition used to define sarcopenia was unusual and the nutrient intake was not monitored.

In conclusion, studies failed to show major improvement in any of the three indicators of sarcopenia in healthy subjects despite a compliance rate over 90%. In frail and sarcopenic older adults, oral nutritional supplementation could improve muscle mass and some parameters of muscle strength and physical performance (Table 1).

#### 3.4.2. Combined Nutritional and Physical Activity/Exercise Interventions

##### Healthy Older Adults

Two studies investigated the impact of exercise combined with nutritional supplementation on muscle mass, muscle strength, and physical performance in healthy Japanese and Italian older women. After a 24-week intervention, whey protein supplementation (22.3 g) ingested twice a week after resistance exercise increased upper and lower limb muscle mass (*p* = 0.029 and < 0.001), handgrip and knee extension strength (*p* = 0.014 and < 0.001), and gait speed (*p* = 0.026) compared to whey protein supplementation alone [29]. An 8-week endurance and resistance exercise program with a daily oral nutritional supplementation (330 kcal, 20 g of proteins, 1.5 g of HMB) improved peak torque isometric knee extension (Δ = 3.32 ± 2.61 Nm; *p* = 0.03), isokinetic strength (Δ = 9.74 ± 3.90 Nm; *p* = 0.02), and 6-min walking test (Δ = 7.67 ± 8.29 m; *p* = 0.04) compared to no treatment [30].

Kirk et al. randomized untrained subjects to a 16-week combined physical and nutritional intervention, consisting of resistance and functional exercise plus a leucine-enriched whey protein supplement to achieve 1.5 g/kg/day of proteins, a physical or nutritional intervention alone, or no treatment [31,32]. A significant improvement in muscle strength and physical performance was noted in combined physical and nutritional groups, and in physical groups alone. However, no differences between the two groups was noted. In other words, leucine-enriched whey protein supplementation did not provide any further benefits, but the authors reported a lack of compliance to supplementation. In a small sample, Markofski et al. demonstrated a beneficial effect of a 12-week daily essential amino acids supplement (15 g) plus supervised endurance exercise on isokinetic leg strength, walking speed and V0_2_ peak in low active subjects [33]. Unfortunately, they did not report nutrient intake. Seino et al. combined a resistance exercise training program with or without a daily oral nutritional supplementation (114 kcal, 10.5 g of proteins, micronutrients: 8.0 mg zinc, 12 μg vitamin B12, 200 μg folic acid, 200 IU vitamin D, and others) in healthy older adults not engaged in an exercise program [34]. After 12 weeks, the combined exercise and nutritional intervention group showed a greater improvement in lean body mass (supplementation effect = 0.63 kg [95%CI: 0.31 to 0.95]) and ASMM (supplementation effect = 0.37 kg [95%CI: 0.16 to 0.58]) but no change in muscle strength and physical performance. However, the authors did not mention the compliance to the interventions. 

##### Older Adults with Low-Protein Intake or Limited Mobility

In Brazilian older adults able to walk independently but with protein a intake <1 g/kg/day, a 3-month resistance training combined with protein supplementation (40 g/day) improved handgrip strength (22.55 ± 6.31 vs. 16.46 ± 3.78 kg, *p* < 0.001), and 5-time sit-to-stand test (11.89 ± 2.87 vs. 17.05 ± 7.69 s, *p* = 0.016) compared to a control group keeping to their daily routine [35]. Bonnefoy et al. followed French older adults at risk of becoming frail but able to walk independently over 4 months [36]. A daily self-administered mobility, strength, balance, and endurance exercise program combined with protein supplementation (10 g/day) had benefits on walking speed in 44% of participants considered to be good compliers. A supervised endurance, resistance, balance, and flexibility exercise combined with a daily oral nutritional supplement (150 kcal, 20 g of whey protein, 800 UI vitamin D) was tested in a large sample of mobility-limited Swedish older adults with vitamin D insufficiency. The control group received the same training program plus a placebo [37,38]. After 6 months, both groups improved knee extensor strength (mean change: 7.27 Nm (95%CI 3.16 to 11.37) for intervention and 9.08 Nm (95%CI 5.03 to 13.14) for placebo; all *p* < 0.001) and physical performance (mean change in 400-m walk speed: 0.08 m/s (95%CI 0.05 to 0.10) for intervention and 0.11 m/s (95%CI 0.08 to 0.14) for placebo; mean increase in SPPB score: 2.1 and 2.6 units respectively; all *p* < 0.05). Interestingly, the authors did not demonstrate differences between groups, suggesting that the nutritional supplementation did not provide additional benefits, but that the interpretation is limited by the lack of data on total caloric and protein intake. 

##### Frail Older Adults

Two studies were performed in frail Japanese older adults. Ikeda et al. conducted a randomized crossover trial among subjects requiring long-term care [39]. A combination of resistance, endurance, and balance training plus nutritional interventions (6 g of branched chain amino acids (BCAA) or 6 g of maltodextrin, 2×/week, before exercise) was carried out twice a week during periods A (3 months) and B (3 months). During the wash-out period of 1 month, participants were engaged in an exercise intervention alone. In spite of a drop-out rate of 21%, the authors showed a significant 10% improvement in lower limb isometric strength and dynamic balance ability in the BCAA group compared to the control group. Yamada et al. demonstrated that a pedometer-based walking intervention with an increment in daily steps of 10% each month increased skeletal muscle index by 0.64% after 6 months [40]. With the adjunction of a daily oral nutritional supplementation (200 kcal, 10 g of protein, 12.5 μg of vitamin D), the increase was 3.16%. These results were significant compared to the control group (*p* = 0.005).

##### Sarcopenic Older Adults

The effects of combined nutritional and exercise interventions have been tested several times in sarcopenic older adults living in the community. Bjorkman et al. randomized 218 Finnish older subjects to 2 × 20 g/day of whey protein supplementation or isocaloric placebo or control with no supplementation [41]. All participants were given instructions on low intensity home-based exercise, the importance of dietary protein intake, and vitamin D supplementation of 20 µg/day. After 12 months, the whey supplementation combined with the home-based program did not enhance muscle mass, muscle strength, and physical performance but a compliance of only 45% was reported. In China, patients undergoing an endurance and resistance exercise training program over 12 weeks, with or without daily nutritional supplementation (231 kcal, 8.61 g of protein), significantly improved ASMMI (mean change: 0.11 kg/m^2^, (95%CI 0.03 to 0.19) vs. −0.21 kg/m^2^, (95%CI −0.43 to 0.02); *p* < 0.05), leg extensors strength (mean change: 3.73 kg, (95%CI 2.28 to 5.18) vs. −0.62 kg/m^2^, (95%CI −2.17 to 0.92); *p* < 0.05) and 5-time sit-to-stand test (mean change: −3.77 kg, (95%CI −4.76 to −2.77) vs. −1.49 kg, (95%CI −2.55 to −0.43); *p* < 0.05) [42]. The beneficial effects of combined nutritional and exercise interventions on sarcopenia parameters were also confirmed in Japanese older women in a 3-month resistance and balance exercise plus daily 6 g amino acid supplementation [43]. In older German men with sarcopenic obesity, whole body electrostimulation plus protein intake of 1.7–1.8 kg/day and 800 UI vitamin D over 16 weeks improved muscle mass, muscle strength, and physical performance as compared to no treatment [44,45]. The skeletal muscle index mean change was 0.018 (95%CI 0.011 to 0.026) vs. −0.008 (95%CI −0.001 to −0.016) (*p* ≤ 0.009). The mean difference for ASMM was 0.45 kg (95%CI 0.26 to 0.65), for maximum dynamic strength “leg press” 155 N (95%CI 73 to 238), and for 10-m gait velocity 0.041 m/s (95%CI 0.020 to 0.063) (all *p* < 0.001). Finally, Kim et al. demonstrated a greater change in 11 m gait velocity for resistance and endurance exercise combined with leucine nutrition supplementation (3 g/day) than for the control group in women with sarcopenic obesity [46].

##### Summary

The literature showed that combined nutritional and physical activity/exercise interventions are efficient to prevent sarcopenia in healthy community-dwelling people, and to improve muscle mass, muscle strength and physical performance in frail and sarcopenic subjects. However, it is important to note that the definition of sarcopenia for the inclusion of the participants varied between studies (Table 2).

#### 3.4.3. Combined Nutritional and Testosterone Therapy Interventions

Two authors studied the effects of nutritional intervention combined with testosterone therapy. Bhasin et al. randomized older men with mobility limitations and daily protein intake < 0.83 g/kg/day to a 6-month controlled diet with weekly placebo injections plus 0.8 g/kg/day of protein, weekly placebo injections plus 1.3 g/kg/day of protein, weekly testosterone injections plus 0.8 g/kg/day of protein, or weekly testosterone injections plus 1.3 g/kg/day of protein [47]. Compared to placebo, testosterone was associated with a greater change in lean body mass (effect size: 3.54 kg, 95%CI 2.88–4.20, *p* < 0.001), ASSM (1.86 kg, 95%CI 1.48-2.23; *p* < 0.001), maximal leg press strength (84.1 N; 95%CI, 7.5–160.8, *p* = 0.03), and chest press strength (37.0 N, 95%CI 18.8–55.1, *p*< 0.001), while protein intake did not influence the positive anabolic response to testosterone therapy. In a smaller study, Visvanathan et al. tested a daily oral nutritional supplement (500–800 kcal) combined with oral testosterone in undernourished older subjects, for one year [48]. The authors failed to demonstrate any effect of this intervention on muscle mass, muscle strength and hospital admissions. However, this study had a high dropout rate of 36% and the total nutrient intake was not assessed.

In community-dwelling older adults, the benefits of testosterone therapy added to nutritional supplementation are open to debate and the maintenance of muscle mass and strength over time after stopping testosterone therapy remains to be demonstrated (Table 3).

#### 3.4.4. Multimodal Interventions (>2 Interventions)

Deer et al. randomized older adults admitted to hospital for an acute medical illness to one of five interventions groups: (1) whey protein supplementation, 40 g/day, (2) in-home resistance training program, (3) combined whey protein supplementation plus exercise program, (4) single testosterone injection, (5) isocaloric placebo [49]. Patients were included at discharge and followed over 4 weeks. Physical performance was improved in all active intervention groups compared to placebo (*p* < 0.05) with no difference between the intervention groups. Readmission rates were highest in the groups receiving isocaloric placebo (28%), followed by the exercise program (15%), the whey protein supplementation (12%), and the combined whey protein supplementation plus exercise program (11%), and testosterone (5%). As this was a pilot study, the sample size in each group was small and the statistical power was limited.

Two studies evaluated the impact of multimodal interventions in community-living frail older adults. In a large sample, Romera-Liebana et al. were interested in the impact of supervised exercise, endurance, resistance, flexibility, balance training + daily nutritional supplement (156 kcal, 11.8g protein, for 6 weeks) + memory workshop + medication review [50]. As compared to a control group receiving standard care and after 3- and 18-month follow-up, the authors reported a significant improvement in handgrip strength (2.84 and 2.49 kg, *p* < 0.001) and SPPB score (1.58 and 1.36 points, *p* < 0.001). Nutrient intake was not monitored and compliance to the intervention was not mentioned. Ng et al. randomized subjects to five different 6-month interventions: (1) supervised and home-based resistance and balance exercise; (2) daily oral nutritional supplementation (300 kcal, 12 g of proteins); (3) cognitive training; (4) combined exercise, nutritional and cognitive interventions or; (5) usual care control [51]. As compared to baseline, the multimodal approach (mean change from baseline in frailty score status based on Fried et al. criteria: −0.87, 95%CI −1.16, −0.59, *p* < 0.05) and exercise alone (mean change from baseline in frailty score status: −0.98, 95%CI −1.26, −0.70, *p* < 0.05) were effective in reversing frailty. However, no difference in hospitalization rates were reported.

In community-dwelling older adults, multimodal interventions showed encouraging results on muscle strength and physical performance but there was a controversial impact on hospitalization rates (Table 4).

## 4. Discussion

This systematic review sums up the findings from studies published since 2010 on the impact of nutritional intervention alone or combined with other treatments on muscle mass, muscle strength, and physical performance in community-dwelling older adults. The literature reveals that results are heterogeneous. Nutritional supplementation as a sole intervention of interest could be an interesting means of improving muscle mass, muscle strength, and physical performance in frail and sarcopenic subjects only. Several studies show the additional effect of endurance and resistance exercise to nutritional supplementation, particularly in frail and sarcopenic older adults. The benefits of testosterone therapy added to nutritional supplementation remain controversial. Finally, multimodal interventions show encouraging results on the muscle strength and physical performance of frail subjects. Regarding the second outcome, only three studies reported the hospitalization rate with no significant improvement after interventions.

In older adults with malnutrition or at risk of malnutrition, oral nutritional supplements, providing at least 400 kcal and 30 g of protein/day for a minimum of 1 month, has been recommended by the European Society for Clinical Nutrition and Metabolism (ESPEN) [20]. Oral nutritional supplements improve protein and caloric intake. In older subjects of any nutritional status and from any settings, a previous systematic review showed that high protein oral nutritional supplementation (> 20% energy from protein) improved muscle strength and body composition and decreased the hospitalization rate [52]. In community-dwelling older adults, we found that protein supplementation (25 to 46 g/day) was effective on muscle mass, strength, and physical performance in frail and sarcopenic subjects [27,28]. However, the effects of oral nutritional supplementation were limited in healthy subjects [24,25,26]. Three factors could explain these results: the participants had high protein intake at inclusion, the results may have been confounded by physical activity which was not reported or assessed, and most studies were underpowered (small sample size). These data support the use of oral nutritional supplement in frail and sarcopenic subjects living in the community.

The benefits of regular physical activity are well established in older adults. Physical activity improves cardiorespiratory fitness, body composition, muscle strength, metabolic parameters, bones, and functionality and reduces the risk of mortality, noncommunicable chronic diseases, cognitive decline, falls, and depression [53,54]. The WHO recommends each week at least 150 min of moderate-intensity endurance exercise, two sessions of resistance exercise, and three sessions of balance exercise in people with poor mobility [53]. Recently, a meta-analysis concluded that physical activity alone and nutritional support combined to physical activity were the most effective interventions to decrease frailty [55]. Our systematic review shows similar findings. Nutritional supplementation combined with resistance and/or endurance exercise had interesting effects on muscle mass, muscle strength, and the physical performance of frail and sarcopenic older adults. While the modalities of the interventions varied widely between studies, it seems that a minimum of two sessions of resistance training per week combined to a protein supplementation of a minimum of 6 g/day should be carried out for 12 weeks. However, this systematic review reveals that the effects of such interventions are less clear in healthy, non-frail, untrained, or low active older subjects. These observations are consistent with previously reported outcomes. A recent meta-analysis highlighted that protein supplementation combined with resistance training did not improve muscle mass, muscle strength and physical performance in non-frail community-dwelling older adults [56]. It suggested that protein supplementation plus resistance and/or resistance training could have beneficial effects in specific groups of older adults.

Testosterone therapy has been suggested to counteract age-related effects and improve muscle mass [57]. However, the use of testosterone can be associated with cardio-metabolic disorders and its benefits do not appear to last after the treatment period [58]. In older undernourished subjects and men with mobility limitations and low protein intake, testosterone combined with protein or caloric supplementation over 6 to 12 months did not demonstrate a positive effect on muscle mass, muscle strength, physical function, or hospital admissions [47,48]. As the effects of testosterone are controversial, it does not seem appropriate to recommend this intervention in this population.

Multimodal interventions combine more than two strategy interventions. The aging process is associated with physical, cognitive, and social changes [2]. Therefore, a multidisciplinary approach is interesting and is recommended for older adults [20]. Previous studies have shown the positive effects of multimodal interventions in nursing home residents and in older malnourished people receiving home care [59,60]. The same results were found in frail community-dwelling older people. A multimodal therapy including caloric and protein nutritional supplementation, multicomponent exercises, and cognitive training for 6 to 24 weeks significantly improved muscle strength and physical performance [50,51]. Interestingly, Ng et al. showed that these improvements were still effective more than a year after the end of the intervention [51]. While the authors did not assess muscle mass, multimodal interventions can be recommended in frail and sarcopenic community-dwelling older adults.

Currently, there is a need to optimize the management of older adults living in the community and especially frail and sarcopenic subjects (Figure 2). Older adults should be routinely screened by their general practitioner with a validated screening tool such as Mini Nutritional Assessment (MNA) [20,61]. Recently, the Global Leadership Initiative on Malnutrition (GLIM) has proposed a new definition of malnutrition based on the presence of one phenotypic criterion (unintentional weight loss, low body mass index, or reduced muscle mass) associated with one etiologic criterion (reduced food intake or assimilation, or disease burden/inflammatory condition) [62]. The results of the screening and the assessment of malnutrition should be used to define the therapeutic strategy. Based on this study, a multimodal intervention including at least a combined nutritional and exercise intervention should be implemented. The modalities of such interventions remain to be clearly defined, but pending further studies, the recommendations of the ESPEN and the WHO should be used. According to the ESPEN, older adults at risk of malnutrition or with malnutrition should receive a daily oral nutritional supplement (minimum 400 kcal and 30 g of proteins/day) [20] and should undergo at least 150 min of moderate-intensity endurance exercise and two sessions of resistance exercise each week [53]. Three sessions of balance exercise should also be performed by people with mobility limitations. Other interventions, such as cognitive interventions, can be added according to an individual’s specific health needs. Finally, specific outcomes must be defined to evaluate the impact of the intervention on muscle mass, muscle strength, and physical function. This study highlights the need to standardize the methods used to assess these outcomes. In the new definition of sarcopenia, the EWGSOP has proposed some tools: (a) dual-energy X-ray absorptiometry and bioelectrical impedance analysis for muscle mass; (b) handgrip strength and 5-time sit-to-stand test for muscle strength; and (c) gait speed, SPPB score, TUG and 400 m walk test for physical performance [5]. These tools could be generalized to all community-dwelling older adults.

This systematic review has some limitations. First, the literature search was performed only on Medline. Additional studies and other unpublished studies could have been omitted. Second, only one author (J.M.) assessed the titles, abstracts, and full-texts for eligibility, and abstracted the data from source studies. Potential discrepancies for study selection and data collection between the authors could not be verified. Third, for most studies, nutritional intake and nutritional expenditure by physical activity were not assessed. Although nutritional supplementation is implemented, we do not know if participants met their energy and protein needs. Finally, the results are heterogeneous. The definition used to select the population, the type of interventions, and the methods used to assess outcomes varied between studies, which limited their comparability. The strength of this review is its systematic methodology and the thorough review of the source studies reported in the tables.

## 5. Conclusions

This systematic review highlights the importance of a multimodal approach, including at least a combined nutritional and exercise intervention, to improve muscle mass, muscle strength, and physical performance, in community-dwelling older adults, especially in frail and sarcopenic subjects. Regarding hospitalization and readmission rates, data are limited and inconclusive. Future studies should continue to investigate the effects of such interventions in older adults living in the community. The population should be carefully selected and the type of interventions and the methods used to assess outcomes should be standardized to improve reliability and comparability between studies.

## Figures and Tables

**Figure 1 nutrients-12-02820-f001:**
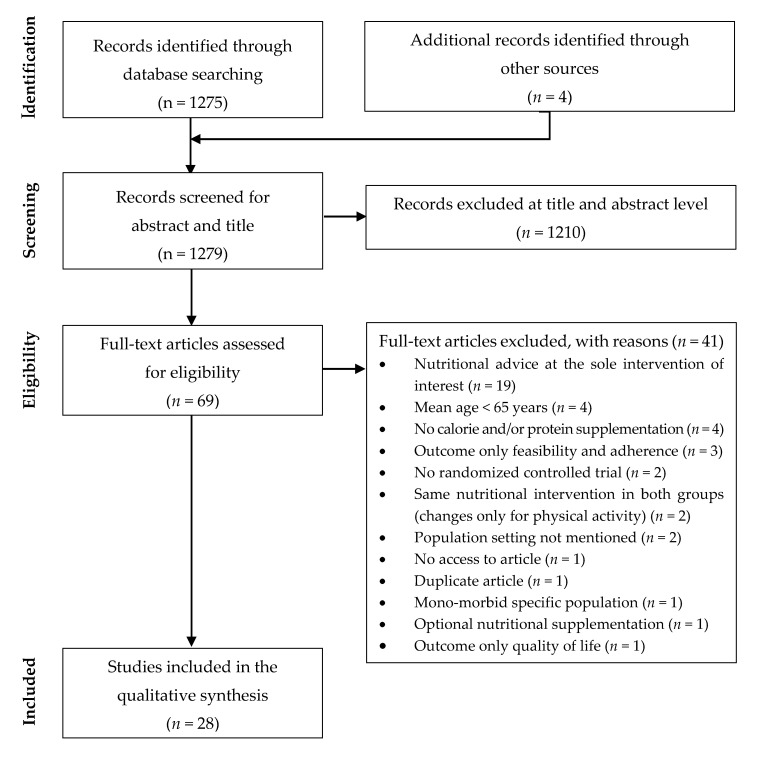
Flow diagram of studies in selection process.

**Figure 2 nutrients-12-02820-f002:**
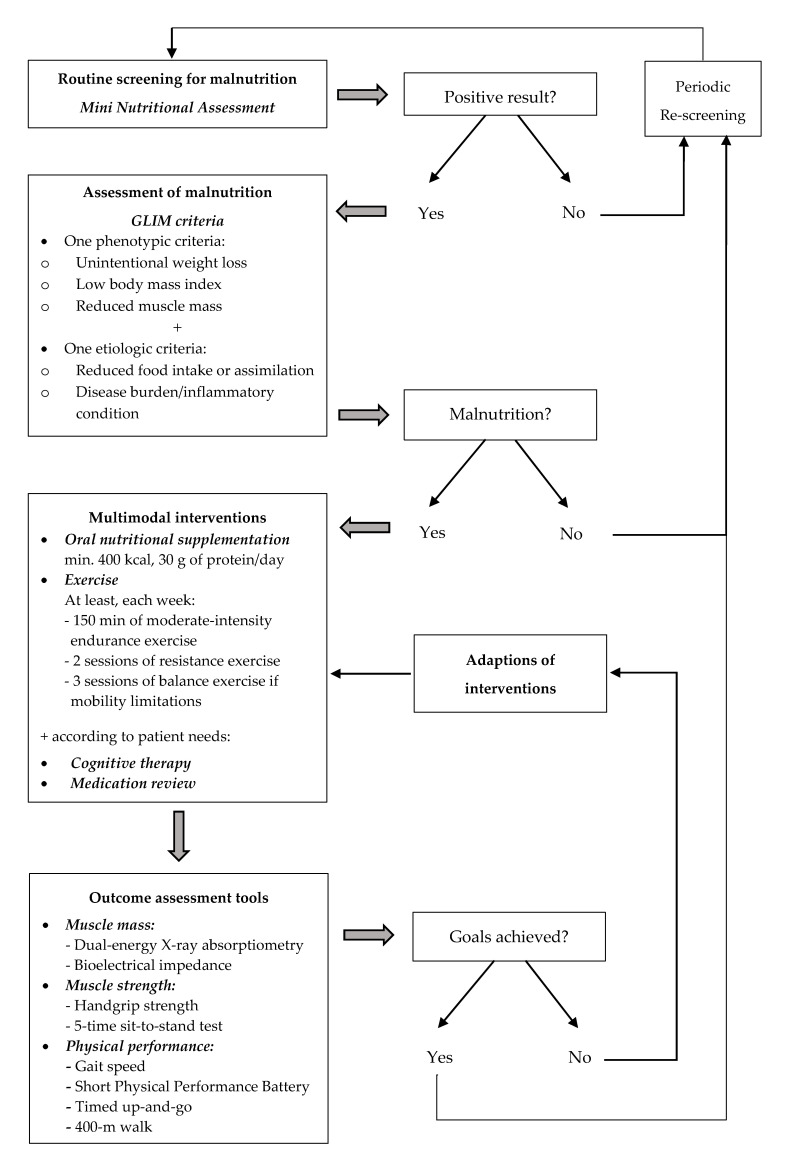
Nutritional management in community-living older adults.

**Table 1 nutrients-12-02820-t001:** Unimodal nutritional interventions in community-dwelling older adults.

Studies	Population	Intervention	Outcomes	Results	Limitations	QS
Ellis et al. 2019 [24]	Healthy men and women 65–89 yrs *n* = 34	6 months INT: amino acid supplement; 3 g HMB, 14g L-arginine, 14 g L-glutamine/day (*n* = 17) CO: isocaloric placebo (*n* = 17)	Lean body mass (DXA) and quadriceps muscle volume (MRI) Physical performance: eight-foot up-and-go test, 25-foot walk test, 12-step timed stair climb	Significant improvement in lean body mass and timed stair climb in INT No change in quadriceps volume and other physical performance parameters	Small sample size, no monitoring of nutrient intake, no assessment of muscle strength	19/28
Zhu et al. 2015 [25]	Healthy women Mean age 74.3 ± 2.7 yrs *n* = 219	2 years INT: protein supplement; 30 g whey, 1×/day (*n* = 109) CO: placebo (*n* = 110)	ASMM (DXA) Muscle strength: handgrip, lower limb muscle strength Physical performance: TUG	No change in ASMM, lower limb muscle strength and TUG between groups Significant decrease in handgrip in INT	High protein intake at inclusion	23/28
Ottestad et al. 2017 [26]	Healthy men and womenHandgrip strength < 20 kg in women, < 30 kg in men Gait speed < 1m/s 5-time sit to stand test ≥ 8.4s ≥ 70 yrs *n* = 50	12 weeks INT: protein-enriched milk; 20 g protein, 2×/day (*n* = 24) CO: isocaloric placebo (*n* = 26)	ASMM (DXA) Muscle strength: handgrip, 1-repetition maximum leg and chest press, 5-time sit-to-stand test Physical performance: stair climbing test (20 steps)	Significant improvement of chest press in INT and CO, but no difference between groups No change in other parameters between groups	High dropout rate in both groups, tests not performed on all subjects	22/28
Kim et al. 2013 [27]	Frail men and women Gait speed < 0.6 m/s MNA < 24 ≥ 65 yrs *n* = 87	12 weeks INT: nutritional supplement; 200 kcal, 12.5 g protein, 2×/day (*n* = 43) CO: no treatment (*n* = 42)	Muscle strength: handgrip Physical performance: one-leg stance SPPB, TUG	No difference between groups in handgrip and one-leg stance SPPB score stable in INT, but decreased in CO Significant improvement of TUG in INT	High dropout rate in INT, no placebo in CO, multiple testing	20/28
Bauer et al. 2015 [28]	Sarcopenic men and women BMI 20–30 kg/m^2^ SPPB score: 4-9 Skeletal muscle mass / body weight × 100: < 37% in men and < 28% in women Mean age 77.7 yrs *n* = 380	13 weeks INT: nutritional supplement; 20 g whey protein, 3 g total leucine, 800 IU vitamin D, 2×/day (*n* = 184) CO: isocaloric placebo (*n* = 196)	ASMM (DXA) Muscle strength: handgrip, 5-time sit-to-stand test Physical performance: SPPB, gait speed (4-m walk), balance test	Significant improvement in ASMM and 5-time sit-to-stand test in INT compared with CO No difference in handgrip, SPPB, gait speed and balance scores between groups	High dropout rate, no monitoring of nutrient intake, definition of sarcopenia not clear	24/28

Abbreviations: ASMM: appendicular skeletal muscle mass, CO: control group, DXA: dual-energy X-ray absorptiometry, HMB: beta-hydroxy-beta-methylbutyrate, INT: Intervention group, MNA: mini nutritional assessment, MRI: magnetic resonance imaging, QS: quality score, BMI: Body mass index, SPPB: short physical performance battery, TUG: timed up-and-go; Yrs: years.

**Table 2 nutrients-12-02820-t002:** Combined nutritional and physical activity/exercise interventions in community-dwelling older adults.

Studies	Population	Intervention	Outcomes	Results	Limitations	QS
Mori et al. 2018 [29]	Healthy women Aged 65–80 yrs *n* = 81	24 weeks *Group A:* supervised and home-based resistance exercise; 2×/week + protein supplement; 22.3 g whey, 2×/week 5 min after exercise (*n* = 27) *Group B:* supervised and home-based resistance exercise; 2×/week (*n* = 27) *Group C:* protein supplement; 22.3 g whey, 2×/week (*n* = 27)	Upper and lower limb muscle mass (BIA) Muscle strength: handgrip, knee extension strength Physical performance: gait speed	Significant improvement in upper and lower limb muscle mass, handgrip strength and gait speed in exercise + protein supplement compared to protein supplement only Significant improvement in lower limb muscle mass and knee extension strength in exercise + protein supplement compared to exercise only and protein supplement only	Compliance to the intervention not reported, characteristics of lost to follow-up not described	19/28
Berton et al. 2015 [30]	Healthy women Mean age 69.5 ± 5.3 yrs *n* = 80	8 weeks Supervised endurance and resistance exercise; 2×/week + INT: nutritional supplement; 330 kcal, 20 g proteins, 1.5 g HMB, 1×/day (*n* = 40) CO: no treatment (*n* = 40)	ASMM (DXA) Muscle strength: isometric knee extension torque, isokinetic strength, handgrip Physical performance: SPPB, 6MWT	No difference in ASMMI, handgrip and SPPB between groups Significant improvement in isometric knee extension torque, isokinetic strength and in 6MWT in INT	No monitoring of nutrient intake, description of exercise training not clear	22/28
Kirk et al. 2020 [31]	Non-frail and untrained men and women Mean age 69 ± 6 yrs *n* = 123	16 weeks *Group A:* no treatment (*n* = 34) *Group B:* supervised resistance and functional exercise; 50 min, 3×/week (*n* = 29) *Group C:* supervised resistance and functional exercise; 50 min, 3×/week + leucine-enriched whey protein supplement; based on individual body-weight 1.5 g/kg/day (*n* = 22) *Group D:* leucine-enriched whey protein supplement; based on individual body weight 1.5 g/kg/day (*n* = 38)	Muscle mass and skeletal muscle index (muscle mass/height^2^) (BIA)	No significant change in muscle mass and skeletal muscle index	Lack of compliance to protein supplement, high dropout rate in protein supplementation group, lack of external validity	17/28
Kirk et al. 2019 [32]	Non-frail and untrained men and women Mean age 68 ± 5 yrs *n* = 51	16 weeks *Group A:* supervised resistance and functional exercise; 50 min, 3×/week (*n* = 29) *Group B:* supervised resistance and functional exercise; 50 min, 3×/week + leucine-enriched whey protein supplement; based on individual body-weight 1.5 g/kg/day (*n* = 22)	Muscle strength: 5-repetition maximum in leg press, chest press, and bicep curl Physical performance: SPPB, 25-m obstacle course, 6MWT	Significant improvement in all parameters in the both groups, with no difference between groups	Lack of compliance to protein supplement, no muscle mass assessment, lack of external validity	16/28
Markofski et al. 2018 [33]	Non-frail independent men and women Low active <7500 steps/day Not engaged in an exercise program Mean age 72 ± 1 yrs *n* = 50	24 weeks *Group A:* nutritional supplement; 15 g essential amino acids, 1×/day + supervised endurance exercise; 50 min, 3×/week (*n* = 14) *Group B:* placebo supplement + supervised endurance exercise; 50 min, 3×/week (*n* = 11) *Group C:* nutritional supplement; 15 g essential amino acids, 1×/day (*n* = 13) *Group D:* placebo (*n* = 12)	Lean body mass (DXA) Muscle strength: isokinetic leg strength Physical performance: 20-m walk, 20-m walk with carry, 400-m walk test, VO2	Compared to baseline: No significant change in lean mass Significant improvement of isokinetic leg strength only in the nutritional supplement + exercise group Significant improvement of walking speed and VO2 peak in both exercise groups, irrespective of supplementation type	No monitoring of nutrient intake, statistical power limited, non-frail not defined, no sample size calculation	19/28
Seino et al. 2018 [34]	Non-disabled men and women Not engaged in an exercise program Mean age 73.5 yrs *n* = 82	12 weeks *Group A:* supervised resistance exercise; 60 min, 2×/week + nutritional supplement; 114 kcal, 10.5 g protein, 1×/day + micronutrient beverage, 1×/day (*n* = 41) *Group B:* supervised resistance exercise; 60 min, 2×/week (*n* = 41)	Lean body mass, ASMM (DXA) Muscle strength: handgrip, knee-extension strength, 5-time sit-to-stand test Physical performance: assessed by single leg stand, gait speed, TUG	Significant improvement in lean body mass and ASMM in exercise and nutritional supplement group compared to exercise only No significant differences between groups in handgrip strength, knee-extension strength, single leg stand, gait speed, TUG, 5-time sit-to-stand test	Few male subjects, compliance to the intervention not mentioned	22/28
De Carvalho Bastone et al. 2020 [35]	Men and women Low protein intake < 1 g/kg/day Able to walk independently Handgrip strength <20 kg in women, <30 kg in men Mean age 75.9 ± 6.7 yrs *n* = 80	3 months *Group A:* home-based supervised progressive resistance exercise program; 60 min, 3×/week (*n* = 20) *Group B:* protein supplement; 40 g protein/day (*n* = 20) *Group C:* combined resistance exercise program and protein supplementation (*n* = 20) *Group D:* daily routine (*n* = 20)	Skeletal muscle index: absolute skeletal muscle mass / height squared (BIA) Muscle strength: handgrip, 30 s and 5-time sit-to-stand test Physical performance: gait speed (8-m course), TUG	Significant improvement in handgrip, gait speed, and 5-time sit-to-stand test in resistance training only and combined resistance plus protein supplement groups compared to control group No significant difference in all parameters between resistance training only and combined resistance plus protein supplement groups	No monitoring of nutrient intake, lack of statistical power	21/28
Bonnefoy et al. 2012 [36]	Men and women Able to walk independently; risk of becoming frail: gait speed < 0.8 m/s ± PASE < 64 for men, < 52 for women Median age 84 yrs *n* = 102	4 months INT: self-administered mobility, strength, balance and endurance exercise program; 20 min, 1×/day + protein supplement; 10 g/day during 1.5 months (*n* = 53) CO: no treatment (*n* = 49)	Fat-free mass (device?) Physical performance: TUG, walking speed, maximum walking time, 1-min sit-to-stand test, six-stair climbing time	Only significant reduction in maximum walking time in control group	Low compliance, protein supplementation only during 1.5 month, evaluators and participants not blinded	17/28
Englund et al. 2017 [37]	Men and Women Mobility limitations: SPPB ≤ 9 Low serum vitamin D level Mean age 77.5 ± 5.4 yrs *n* = 149	6 months *Group A:* supervised endurance, resistance, balance and flexibility exercise; 60 min, 3×/week + nutritional supplement; 150 kcal, 20 g whey protein, 800 UI vitamin D, 1×/day (*n* = 74) Group B: supervised endurance, resistance, balance and flexibility exercise; 60 min, 3×/week + placebo, 30 kcal, 1×/day (*n* = 75)	ASMM (DXA) Muscle strength: isokinetic strength	No improvement in ASMM in both groups Improvement in muscle strength in both groups No significant differences for all parameters between groups	No monitoring of nutrient intake	21/28
Fielding et al. 2017 [38]	Men and Women Mobility limitations: SPPB ≤ 9 Low serum vitamin D level Mean age 77.5 ± 5.4 yrs *n* = 149	6 months *Group A:* supervised endurance, resistance, balance and flexibility exercise; 60 min, 3×/week + nutritional supplement; 150 kcal, 20 g whey protein, 800 UI vitamin D, 1×/day (*n* = 74) Group B: supervised endurance, resistance, balance and flexibility exercise; 60 min, 3×/week + placebo, 30 kcal, 1×/day (*n* = 75)	Physical performance: gait speed (400-m walk capacity), SPPB	Significant improvement in gait speed and SPPB in both groups but no significant difference between groups	No monitoring of nutrient intake	21/28
Ikeda et al. 2016 [39]	Pre-frail and frail men and women according to Fried et al. Mean age 78.4 ± 7.8 yrs and 80.4 ± 8.9 yrs in the 2 groups *n* = 52	Cross-over design: two time 3 months of supplementation combined with exercise, washout of 1 month with exercise only Supplementation: 6g of BCAA or 6 g of maltodextrin, 2×/week 10 min before exercise Exercise: supervised resistance, endurance, balance exercise; 2×/week	Muscle strength: handgrip, upper and lower limb isometric strength Physical performance: TUG, dynamic balance ability	Significant improvement in lower limb isometric strength and dynamic balance ability in BCAA group compared to the control group after crossover	High dropout rate, no monitoring of nutrient intake, population including both pre-frail and frail subjects	19/28
Yamada et al. 2015 [40]	Non-frail and frail (Cardiovascular Health Study criteria) men and women Able to walk independently Age ≥ 65 yrs *n* = 227	6 months *Group A:* pedometer-based walking program; increasing of daily steps by 10% each month with ankle weight 0.5 kg + nutritional supplement; 200 kcal, 10 g protein, 12.5 ug vitamin D, 1×/day (*n* = 79) *Group B:* pedometer-based walking program; increasing of daily steps by 10% each month with ankle weight 0.5 kg (*n* = 71) *Group C:* daily routine (*n* = 77)	Skeletal muscle mass index: muscle mass/height^2^ (BIA)	Significant improvement of skeletal muscle index in exercise + nutrition and exercise alone compared to control group Effects more pronounced in the subgroup of frail subjects	No monitoring of nutrient intake, compliance and number of steps in both groups before and after intervention not reported	20/28
Bjorkman et al. 2020 [41]	Sarcopenic men and women Able to walk independently Handgrip <20 kg in women, < 30 kg in men or gait speed <0.80 m/s 75–96 yrs *n* = 218	12 months Instructions on low-intensity home-based exercise, importance of dietary protein and vitamin D supplementation 20 µg/day + *Group A:* no treatment (*n* = 72) *Group B:* isocaloric placebo (*n* = 73) *Group C:* protein supplement; 20g whey protein, 2×/day (*n* = 73)	Skeletal muscle index: skeletal muscle mass / height^2^ (BIA) Muscle strength: handgrip physical performance: SPPB	No significant differences in skeletal muscle index and physical performance between groups Significant reduction in muscle strength in all groups	Dropout higher in control group compared to other groups, low compliance and adherence in intervention groups	21/28
Zhu et al. 2019 [42]	Sarcopenic men and women Sarcopenia: Asian Working Group criteria ≥ 65 yrs *n* = 113	12 weeks *Group A:* supervised endurance and resistance exercise; 45 min, 1×/week + home session, resistance exercise, min 1×/week (*n* = 40) *Group B:* supervised endurance and resistance exercise; 45 min, 1×/week + home session, resistance exercise, min 1×/week nutritional supplement + oral nutritional supplement; 231 kcal, 8.61 g protein, 1×/day (*n* = 36) *Group C:* daily routine (*n* = 37)	ASSM (DXA) Muscle strength: handgrip strength, leg extensors strength, 5-time sit-to-stand test Physical performance: gait speed (6-m walk test)	Compared to control group: Significant improvement of ASMMI in exercise + nutritional supplement group only Significant improvement of leg extensors strength and 5-time sit-to-stand test in both intervention groups No significant improvement of handgrip and gait speed	No monitoring of nutrient intake, high dropout rate	21/28
Kim et al. 2012 [43]	Sarcopenic women Sarcopenia: ASMMI < 6.42 kg/m^2^, knee extension strength < 1.01 Nm/kg, gait speed < 1.22 m/s, BMI < 22.0 kg/m^2^ ≥ 75 yrs *n* = 155	3 months *Group A:* supervised resistance and balance exercise; 60 min, 2×/week + protein supplement; 3.0 g of amino acid, 2×/day (*n* = 38) *Group B:* supervised resistance and balance exercise; 60 min, 2×/week (*n* = 39) *Group C:* protein supplement; 3.0 g of amino acid, 2×/day (*n* = 39) *Group D:* health education (*n* = 39)	Leg muscle mass (BIA) Muscle strength: knee extension strength Physical performance: walking speed	Compared to health education group: Significant increase in leg muscle mass and muscle strength only in the exercise + nutrition group Signification increase of walking speed in exercise + nutrition group and exercise only groups	Multiple testing, no monitoring of nutrient intake	21/28
Kemmler et al. 2018 [44]	Sarcopenic and obese men Sarcopenia: ASSM/BMI <0.789 Obesity: fat mass > 27% ≥ 70 yrs *n* = 67	16 weeks *INT:* whole-body electro-myostimulation; 14 to 20 min, 1.5×/week + whey protein (aim: protein intake 1.7–1.8 g/kg/day) and 800 UI vitamin D/day (*n* = 33) *CO:* no treatment (*n* = 34)	Muscle distribution of intra-fascial fat-free muscle volume of the mid-thigh (MRI) and ASMM (BIA) Muscle strength: leg-extensor strength Physical performance: 10-m gait velocity	Significant improvement of all parameters in intervention group compared to baseline and compared to control group	Sarcopenia and obesity not defined according usual definitions, lower protein intake than prescribed in intervention group, high MRI assessment refusal rate	19/28
Kemmler et al. 2017 [45]	Sarcopenic and obese men Sarcopenia: ASSM/BMI < 0.789 Obesity: fat mass > 27% ≥ 70 yrs *n* = 100	16 weeks *Group A:* whole-body electro-myostimulation; 14 to 20 min, 1.5×/week + whey protein (aim: protein intake 1.7–1.8 g/kg/day) and 800 UI vitamin D/day (*n* = 33) *Group B:* whey protein (aim: protein intake 1.7–1.8 g/kg/day) and 800 UI vitamin D/day (*n* = 33) *Group C:* no treatment (*n* = 34)	Skeletal muscle index: ASSM/BMI (BIA) Muscle strength: handgrip	Significant improvement of skeletal muscle index in the 2 intervention groups Significant increase in handgrip strength with electromyostimulation only	Sarcopenia and obesity not defined according usual definitions, pro tein intake lower than prescribed in both protein-supplemented group	19/28
Kim et al. 2016 [46]	Sarcopenic and obese women Sarcopenia: skeletal muscle index < 5.67 kg/m2, handgrip < 17 kg, gait speed < 1.0 m/s Obesity: fat mass > 32% > 70 yrs *n* = 139	3 months *Group A:* supervised resistance and endurance exercise; 60 min, 2×/week + nutritional supplement 3.0 g leucine, 20 mg vitamin D and 350 mL of tea catechin fortified, 1×/day (*n* = 36) *Group B:* supervised resistance and endurance exercise; 60 min, 2×/week (*n* = 36) *Group C:* nutritional supplement 3.0 g leucine, 20 mg vitamin D and 350 mL of tea catechin fortified, 1×/day (*n* = 34) *Group D:* health education (*n* = 34)	ASMM (BIA) Muscle strength: handgrip Physical performance: 11-m gait velocity	No significant changes in ASMM and handgrip strength between the groups Significant improvement of gait velocity in exercise + nutrition group	Multiple testing, no monitoring of nutrient intake	22/28

Abbreviations: ASMM: Appendicular skeletal muscle mass, ASMMI: appendicular skeletal muscle mass index, BCAA: branched chain amino acids, BIA: bioelectrical impedance analysis, BMI: body mass index, CO: control group, DXA: dual-energy X-ray absorptiometry, HMB: beta-Hydroxy beta-methylbutyric acid, INT: intervention group, MRI: magnetic resonance imaging, PASE: physical activity scale for the elderly, SPPB: short physical performance battery, TUG: timed up and go, 6MWT: six-minute walk test; Yrs: years.

**Table 3 nutrients-12-02820-t003:** Combined nutritional and testosterone therapy interventions in community-dwelling older adults.

Studies	Population	Intervention	Outcomes	Results	Limitations	QS
Bhasin et al. 2018 [47]	Men Mobility limitations: SPPB 3–10 Daily protein intake < 0.83 g/kg/day Mean age 73 ± 5.8 yrs *n* = 92	6 months *Group A:* placebo injections weekly + 0.8 g/kg/day protein (n = 24) *Group B:* placebo injections weekly + 1.3 g/kg/day protein (*n* = 24) *Group C:* testosterone enanthate 100 mg intramuscularly weekly + 0.8 g/kg/day protein (*n* = 22) *Group D:* testosterone enanthate 100 mg intramuscularly weekly + 1.3 g/kg/day protein (*n* = 22)	Lean body mass, ASSM (DXA) Muscle strength: maximal leg press and chest press strength Physical performance: 6MWT, stair climbing, 50-m walk carrying a load equaling 20% body mass	Regardless of whether patients received testosterone or placebo: no change in muscle mass, muscle strength, and physical function between men assigned to 0.8 vs 1.3 g/kg/d of protein Regardless of whether patients received 0.8 vs 1.3 g/kg/d of protein: change in lean mass, ASSM and muscle strength but not in physical function in men randomized to testosterone compared to placebo	No combined physical activity, only men, pre-packaged controlled meals (not representative of reality), statistical power limited	24/28
Visvanathan et al. [48] 2016	Undernourished men and women MNA: 17 and 23.5 BMI < 22 kg/m2 or self-reported weight loss ≥7.5% in the last 3 months Aged ≥ 65 yrs *n* = 53	12 months INT: nutritional supplement; 500–800 kcal, 1×/day + oral testosterone; undecanoate 40 mg/day women, 160 mg/day men (*n* = 25) CO: nutritional supplement; 35–45 kcal, 1×/day + placebo medication (*n* = 26)	Muscle mass (BIA) Muscle strength: handgrip Hospital admissions	No significant difference in all parameters in each arm and between the treatment arms	Expected sample size not reached, high dropout rate, no monitoring of nutrient intake	22/28

Abbreviations: BMI: body mass index, CO: control group, DXA: dual-energy X-ray absorptiometry, INT: intervention group, MNA: nini nutritional assessment, SPPB: short physical performance battery, Yrs: years, 6MWT: six-minute walk test.

**Table 4 nutrients-12-02820-t004:** Multimodal interventions in community-dwelling older adults.

Studies	Population	Intervention	Outcomes	Results	Limitations	QS
Deer et al. 2019 [49]	Men and women admitted to hospital for an acute medical illness Residing at home before and after hospitalization Able to walk across a small room 2 weeks before hospitalization and to stand independently Mean age 78.1 ± 8.0 yrs *n* = 100	During 4 weeks after discharge *Group A:* protein supplement; 20 g whey protein, 2×/day (*n* = 20) *Group B:* in-home resistance training program, 3×/week + placebo supplement (*n* = 21) *Group C:* combined protein supplementation + resistance training (*n* = 20) *Group D:* single testosterone injection; enanthate 100 mg for women, 200 mg for men (*n* = 19) *Group E*: placebo (*n* = 20)	Lean body mass and ASMM (DXA) Physical performance: SPPB 30-day readmission	Significant improvement in SPPB score in all active intervention groups compared to placebo No significant differences in body composition between groups Readmission rates highest in placebo (28%), followed by exercise + placebo (15%), protein supplementation (12%), exercise + protein supplementation (11%) and testosterone (5%)	Pilot study with many interventions studied in the same trial, statistical power limited, no monitoring of nutrient intake	20/28
Romera-Liebana et al. 2017 [50]	Prefrail and frail men and women TUG: 10–20s No severe cognitive impairment Mean age 77.3 yrs *n* = 352	Follow-up at 3 and 18 months INT: multimodal therapy; supervised exercise, endurance, resistance, flexibility, balance training; 60min, 2×/week, for 6 weeks + nutritional supplement; 156 kcal, 11.8g protein, 1×/day, for 6 weeks + memory workshop; 90 min, 2×/week for 6 weeks + medication review (*n* = 176) CO: standard care (*n* = 176)	Muscle strength: handgrip strength Physical performance: SPPB, standing balance, stretching and unipodal station test	Significant improvement in all parameters in the intervention group at 3 and 18 months compared to control group	No monitoring of nutrient intake, no assessment of muscle mass, high dropout rate at 18 months, compliance to the intervention not mentioned	22/28
Ng et al. 2015 [51]	Pre-frail and frail men and women according to Fried et al. Able to walk independently Mean age 70.0 ± 4.7 yrs *n* = 246	24 weeks *Group A*: resistance and balance exercise; supervised 12 weeks, 90 min, 2×/week and home-based, 12 weeks, 90 min, 2×/week (*n* = 48) *Group B:* nutritional supplement; 300 kcal, 12 g proteins, 1×/day (*n* = 49) *Group C:* cognitive training; 2h, 1×/week (*n* = 50) *Group D:* combined intervention (exercise, nutritional supplement and cognitive training) (*n* = 49) *Group E:* standard care (*n* = 50)	Frailty score status: unintentional weight, slowness (6m fast gait speed), weakness (knee extension strength), exhaustion (SF-12 scale) and low-activity (longitudinal ageing physical activity questionnaire) Hospitalization rate: self-reported	Significant reduction of frailty score in exercise and combined intervention groups only compared to baseline No difference in hospitalization rate	No monitoring of nutrient intake, population including both pre-frail and frail subjects	23/28

Abbreviations: ASMM: appendicular skeletal muscle mass, CO: control group, DXA: dual-energy X-ray absorptiometry, INT: intervention group, SPPB: short physical performance battery, TUG: timed up-and-go, Yrs: years.

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
