# Peer review of "Nutritional Intervention to Prevent the Functional Decline in Community-Dwelling Older Adults: A Systematic Review"

_nutrients, 2020, doi:10.3390/nu12092820_

Round 1

Reviewer 1 Report

Thank you for the opportunity to review this manuscript. Overall, the manuscript was well written. I just have some minor suggestions as follows:

  1. Figure 2: should be renamed as "Management of nutrition in community-living older adults", or "Nutritional management in community-living older adults"
  2. Reference: I think it's worth to include this paper in the Discussion: Negm AM, Kennedy CC, Thabane L, et al. Management of Frailty: A Systematic Review and Network Meta-analysis of Randomized Controlled Trials. J Am Med Dir Assoc. 2019;20(10):1190-1198. doi:10.1016/j.jamda.2019.08.009

Reviewer 2 Report

The topic of verifying the effectiveness of dietary interventions alone or combined with others among older people is timely. However, this is complicated due to the multitude of data and their varying quality.

In this manuscript, the authors presented a systematic review in accordance with PRISMA recommendations. The limitation of the research is that the publication search was carried out in one scientific database and only one of the authors evaluated the titles, abstracts and full texts and extracted data from source publications. This may have resulted in some relevant publications being omitted. However, out of 1279 publications, 28 were finally selected according to the eligibility criteria for analysis. This seems to be enough articles to draw important conclusions. It is appropriate that the authors have evaluated the studies in terms of their quality according to the criteria proposed by Downs and Black. Most studies in this review were qualified as ”good” and only two studies as ”excellent”. This indicates the need to continue research of high scientific quality among older people.

In this systematic review, a heterogenous population of healthy, mobility-limited, frail and sarcopenic elderly people was assessed. This approach is appropriate because it gives the possibility to compare nutritional interventions depending on the patient's condition. In general, I have a positive opinion of the manuscript. Conclusions are drawn with caution, underlining the need for multi-factorial intervention and setting out actions for the future. The authors make recommendations which should be taken to increase the quality of research.

I have some reservations about the title, which indicates ”nutritional care”, which does not fully reflect the content of the paper. Please rethink it in terms of "nutritional intervention" or "multimodal intervention". In general, the manuscript is edited very carefully. Minor errors to be corrected are in the discussion: line 361, I think it should be "use" and not "used"; in line 379, "related" is used twice. Figure 2 shows the GLIM criteria in lower case, written Glim.
